# Few-Round Learning for Federated Learning

**Younghyun Park**[*]
dnffkf369@kaist.ac.kr

**Dong-Jun Han**[*]
djhan93@kaist.ac.kr

**Do-Yeon Kim**
dy.kim@kaist.ac.kr

**Jun Seo**
tjwns0630@kaist.ac.kr

**Jaekyun Moon**
jmoon@kaist.edu

School of Electrical Engineering,
Korea Advanced Institute of Science and Technology (KAIST)

## Abstract

In federated learning (FL), a number of distributed clients targeting the same task collaborate to train a single global model without sharing their data. The learning process typically starts from a randomly initialized or some pretrained model. In this paper, we aim at *designing an initial model* based on which an arbitrary group of clients can obtain a global model for its own purpose, within only a few rounds of FL. The key challenge here is that the downstream tasks for which the pretrained model will be used are generally unknown when the initial model is prepared. Our idea is to take a meta-learning approach to construct the initial model so that any group with a possibly unseen task can obtain a high-accuracy global model within only *R rounds of FL*. Our meta-learning itself could be done via federated learning among willing participants and is based on an episodic arrangement to mimic the $R$ rounds of FL followed by inference in each episode. Extensive experimental results show that our method generalizes well for arbitrary groups of clients and provides large performance improvements given the same overall communication/computation resources, compared to other baselines relying on known pretraining methods.

## 1 Introduction

Today, valuable data are being collected increasingly at distributed edge nodes such as mobile phones, wearable client devices and smart vehicles/drones. Directly sending these local data to the central server for model training raises significant privacy concerns. To address this issue, an emerging trend known as federated learning (FL) [13, 9, 1, 11, 20, 16, 15], where server uploading of local data is not necessary, has been actively researched. In FL, a large group of distributed clients interested in solving the same task (e.g., classification on given categories of images) collaborate in training a single global model without sharing their data. While standard supervised learning uses some dataset $D$ to find the model $\phi$ that would minimize a loss function $f(\phi, D)$, FL in comparison seeks the model $\phi$ that minimizes the averaged version of the local losses $f(\phi, D_k)$, computed at each node $k$ using local data $D_k$. The learning process typically starts from a randomly initialized or some pretrained model and is carried out through iterative aggregation of the local model updates.

### 1.1 Backgrounds and Main Contributions

**Motivation.** Unfortunately, FL generally requires a large number of communication rounds between the server and the clients for model exchange, to achieve a desired level of performance. This makes

---

[*]Equal contribution.

35th Conference on Neural Information Processing Systems (NeurIPS 2021).

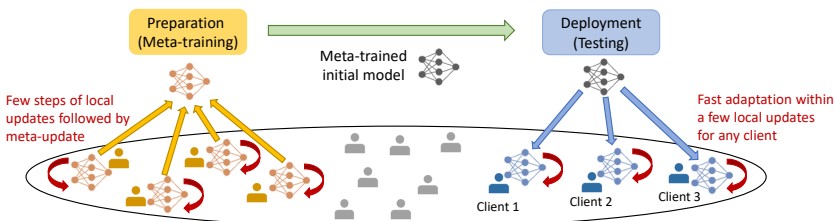

(a) **Personalized FL:** meta-training geared to few steps of local updates at any client.

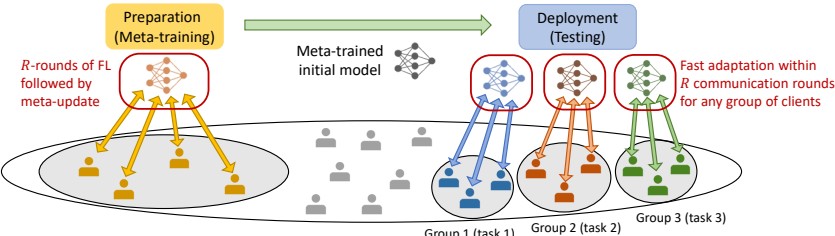

(b) **Proposed few-round FL:** meta-training geared to few rounds of FL at any group.

Figure 1: Basic concepts of personalized FL and proposed few-round learning. In the meta-training phase (or preparation stage) of our scheme, the service provider prepares the initial model with willing participants. Once this preparation is over, the service provider offers this initial model to any group of clients (possibly including the meta-training participants) who hope to perform its own task within few-round FL. In order to facilitate $R$-round FL at deployment, we take an episodic training strategy that mimics actual deployment preceded by $R$ FL rounds in the meta-training phase with a new set of participants chosen in each episode.

the implementation of FL a significant challenge in bandwidth-limited or time-sensitive applications. Especially in real-time applications (e.g., connected vehicles or drones) where the model should quickly adapt to dynamically evolving environments, the requirement on many communication rounds becomes a major bottleneck.

**Goal and challenge.** To tackle this problem from the service provider's perspective, we aim to *prepare an initial model* that can quickly adapt to any group (focusing on its own task) of clients within only a few rounds of FL. The key challenge here is that the task of the group conducting FL (i.e., the downstream task for which the prepared model will be used) is generally not known when the service provider prepares the initial model. In the context of classification, different tasks mean classification involving different sets of classes. For example, classifying diseases $A, B, C$ (task 1) is a different task compared to classifying diseases $D, E, F$ (task 2). Since the group conducting FL for the downstream task can include classes that are unseen during preparation, *existing FL approaches cannot tackle this problem.*

**Key idea.** Our key idea is to adopt meta-learning (which enables reliable prediction even when the task at inference is unseen when the model was meta-trained) to prepare the initial model that enables *few-round FL*. In other words, we aim to meta-train an initial model for few-round downstream FL. Once meta-training is over, the service provider would offer the trained model to some clients who want to solve a common task after collaborating through a quick few rounds of FL. These clients may or may not be the participants of the earlier meta-training phase, and their classification task is generally considered unseen during meta-training. A high-level description of our idea is depicted in Fig. 1(b). Given a small target value $R$, we take an episodic training approach to enable $R$-round FL for any group of clients. In essence, we find the initial model $\phi$ that would minimize the average of local losses $f(\theta^R(\phi), D_k)$, where $\theta^R(\phi)$ is the model to be updated from $\phi$ through $R$ rounds of FL among future clients in the deployment stage. Despite the high practical significance of this problem formulation, to the best of our knowledge, this is the first work to propose a meta-learning strategy geared to *few-round FL*. It is also worth mentioning that model preparation is not a real-time requirement and can often be done when bandwidth demands are sparse.

**Comparison with personalized FL.** We stress that our idea has a different purpose and approach relative to the recent line of works on federated meta-learning [12, 4], which initiate a model for personalized optimizations at local clients (see Fig. 1(a)). The goal of these approaches is to obtain a personalized local model at each client within a few steps of gradient descents, in the deployment

stage. To achieve this goal, in the preparation stage, a few steps of local updates and meta-update are first performed at each participant independently (with its own local data), and FL (or aggregation) is adopted just to take advantage of data of various participants: these approaches seek $\phi$ that minimizes the average of local losses $f(\theta_k(\phi), D_k)$, where $\theta_k(\phi)$ is the local model updated from $\phi$ through a number of gradient steps using local data $D_k$. In contrast to personalized FL that focuses on local client models in the deployment stage, our *few-round learning* inherits the ability of FL at deployment to obtain a *global model*. Hence, for our scheme, it is natural to adopt FL in the preparation stage to mimic the $R$-round FL scenario at deployment; in the preparation stage, meta-update is performed at each participant after the collaborative $R$ FL rounds. To sum, our approach aims to prepare an initial model that leads to a *global model* within a "few rounds of FL", while personalized FL aims for an initial model leading to *personalized models* within "few steps of local updates" based only on the local data. These are obviously two completely different problems with distinct solutions.

**Main contributions.** Technically, we utilize a model-agnostic meta-learning (MAML) approach to prepare the initial model via an episodic training strategy. While directly applying MAML independently to each local model leads to existing solutions on personalized FL [12, 4], in our approach, $R$ rounds of local updates and aggregations are first performed in each episode before the meta-update process. This unique episode construction compared to personalized FL methods mimics the deployment stage where actual inference is preceded by an $R$-round FL procedure. Another key ingredient in our solution is to adopt *prototype aggregation* in each FL round to construct global prototypes that serve as better class representatives compared to the locally computed prototypes, in learning embedding space. This strategy is especially effective when a non-IID (independent, identically distributed) data distribution across clients tends to induce a significantly biased model after performing local updates. The global prototypes serve as prior knowledge, a form of regularization, and prevent local models from overfitting to the local data. Moreover, the global prototypes (reflecting all classes across clients) can assist the local models to learn a more general embedding space. We call this approach a global prototype-assisted learning (GPAL) strategy. Our main contributions are summarized as follows:

- We formulate **a new problem of high practical significance**, namely, **few-round learning**, where the goal is to prepare an initial model that can quickly adapt to any group of clients within only a few rounds of FL.
- We propose a **meta-training algorithm** specifically **geared to $R$ rounds of FL** followed by inference, to be performed by a group of clients on a possibly unseen task.
- We **guarantee convergence** of our meta-training algorithm via theoretical analysis.
- We show via experiments that our scheme **outperforms existing pretraining approaches** including fine-tuning via FedAvg and personalized FL in both IID and non-IID scenarios.

## 1.2 Related Works

**Few-shot learning.** Few-shot learning is an instantiation of meta-learning. In the context of image classification, few-shot learning typically involves episodic training where each episode of training data is arranged into a few training (support) sample images and validation (query) samples to mimic inference that uses only a few examples [19]. Through a repetitive exposure to a series of varying episodes with different sets of image classes, the model learns to handle new tasks (classification against unseen classes) each time. Two widely-known few-shot learning methods with different philosophical twists, which are also conceptually relevant to the present work, are MAML [5] and Prototypical Networks [18]. MAML attempts to generate an initial model from which different models targeting different tasks can be obtained quickly via just a few gradient updates. The idea is that the initial model is learned via meta-training to develop an internal representation that is close in some sense to a variety of unseen tasks. Prototypical Networks, on the other hand, learn embedding space such that model outputs cluster around class prototypes, the class-specific centroids of the embedder outputs. With episodic training, simple Prototypical Networks are surprisingly effective in learning inductive bias for successful generalization to new tasks.

We stress that our few-round learning scheme (that targets a few global rounds of FL) has different purpose and technical approach compared to the existing works on few-shot learning (that targets few shots of data sample). Nevertheless, we take advantage of both concepts on MAML and Prototypical Networks to achieve our own goal: we adopt MAML in updating the initial model specifically geared to $R$-round FL, and adopt both *prototype aggregation* and *prototype-assisted learning* strategies to learn a general embedding space and successfully handle the non-IID issue in FL.

**Federated meta-learning.** Recent research activity has focused on improving model personalization via federated meta-learning [12, 3, 4, 7]. The common goal of these works is to generate an initial model based on which each new client can find its own optimized model via a few local gradient steps and using only its own data. In these works, meta-learning employed during federated learning intends to enable each client to handle previously unseen tasks, in the spirit of MAML of [5]. User-specific next-word prediction at individual smartphones, for example, is a possible application. Compared to this line of work, we focus on creating an initial model that leads to a high-accuracy *global model*, rather than personalized models. In this way, we seek to take advantage of a higher variety of data as well as the larger data volume that would be made available through collaborative learning of a group of distributed nodes. A clear example is the diagnosis of a broader class of diseases that would be possible through collaborative training across more examples contributed by a larger group of individuals. Personalized FL methods (e.g., [12, 4]) especially have disadvantage in non-IID settings where each client necessarily lacks a sufficient variety of data. The results are reported in Section 4.

**One-shot FL.** Another line of work recently focused on one-shot FL, where the goal is to train a global model with just one communication round between the server and the clients. The authors of [6] proposed an ensemble method to choose reliable client-specific models from given clients. In the work of [17], local clients send XOR-encoded MNIST image data to the server, and the server decodes it to train the global model. While the server would need certain data in advance to decode the received results, XOR operation can serve as data augmentation while preserving privacy. In the fusion learning of [8], each local client uploads both the model parameters and the distribution parameters to the server. The server generates artificial data samples from the distribution parameters to train a global model. When the data gets complex, however, it is not clear whether conversion into a simple distribution would be reliable. Compared to the existing works on one-shot FL that employ some randomly initialized model, the key difference of our method is the use of *meta-learning* to prepare an initial model which can adapt to unseen tasks of individual groups' of clients within $R$ rounds of FL. The advantage of our scheme compared to these methods is shown in Section 4.

## 2 Proposed Few-Round Learning Algorithm

### 2.1 Problem Setup

**Federated learning.** Let $N$ be the number of clients in the system. FL allows each distributed node $k$ with a dataset $D_k$ to participate in iterative learning of a global model $\theta$ without having to reveal its data to anyone else including the central server. As a given round $r$ starts, each of $K$ participating nodes (generally chosen anew every round) downloads a global model $\theta^r$ from the server and updates it using its own local data $D_k$. The updated local models $\theta_k^{r+1}$ get all uploaded to the server to be aggregated to a new model $\theta^{r+1} = \sum_{k=1}^{K} \mu_k \theta_k^{r+1}$, according to the relative dataset sizes $\mu_k = \frac{|D_k|}{\sum_{j=1}^{K} |D_j|}$. The same process gets repeated. FL generally requires a significant number of such global rounds to achieve the desired accuracy, with each round taking up substantial communication resources.

**Problem formulation.** In preparing an initial model $\phi$ for any group of clients to pursue a few FL rounds, we use meta-learning based on episodic training, where each episode is constructed to mimic $R$ FL rounds followed by inference. Once meta-training is over, in the deployment phase, the service provider offers the trained initial model $\phi$ to any group of clients wishing to pursue inference on some common task (possibly unseen during meta-training) after collaborating for $R$ rounds of FL.

### 2.2 Meta-Training (Preparation Stage)

More precisely stated, our meta-training phase is to find $\phi$ that minimizes the objective function

$$F(\phi) = \mathbb{E}_{A_t \sim p(\mathcal{A})} \left[ \frac{1}{K} \sum_{k \in A_t} f(\theta^R(\phi), D_k) \right] \tag{1}$$

where $A_t$ is a specific group with $K$ participants drawn from $p(\mathcal{A})$, the distribution over all possible groups, each with $K$ participants; $\theta^R(\phi)$ is the model after $R$ rounds of FL in group $A_t$, starting from $\phi$; and $D_k$ is the local dataset of participant $k$ in group $A_t$. In comparison, the objective function for personalized FL methods (e.g., Per-FedAvg of [4]) is $F(\phi) = \frac{1}{N} \sum_{k=1}^{N} f(\theta_k(\phi), D_k)$ where $N$ is the number of clients in the system and $\theta_k(\phi)$ is the model after a few gradient steps at client $k$ starting from $\phi$. We also reiterate that conventional FL aims at minimizing $F(\phi) = \frac{1}{N} \sum_{k=1}^{N} f(\phi, D_k)$.

---

**Algorithm 1** Proposed Meta-Training Algorithm for Few-Round Learning

---

**Input:** Initialized model $\phi^0$ **Output:** Model $\phi^T$ after $T$ training episodes

1: **for** each training episode $t = 0, 1, ..., T-1$ **do**
2:     The server constructs a group $A_t \sim p(A)$ of $K$ participants chosen out of $N$ users.
3:     Each participant $k \in A_t$ splits $D_k$ into support set $S_k$ and query set $Q_k$.
4:     $\theta^0 \leftarrow \phi^t$
5:     **for** each communication round $r = 0, 1, ..., R-1$ **do**
6:         **for** each participant $k$ **in parallel do**
7:             Download $\theta^r$ and $\Gamma^{r-1}$ from the server (download only $\theta^r$ when $r = 0$)
8:             **for** each class $c \in C_k$ **do**
9:                 $\Gamma_k^r(c) = \frac{1}{|S_k(c)|} \sum_{x \in S_k(c)} g_{\theta^r}(x)$      // Local prototype calculation with support set $S_k$
10:             **end for**
11:             $\theta_k^{r+1} \leftarrow \theta^r - \alpha \nabla_{\theta^r} f(\theta^r, S_k)$      // Local update of $\theta$ with support set $S_k$ and GPAL
12:         **end for**
13:         $\theta^{r+1} = \sum_{k=1}^K \lambda_k \theta_k^{r+1}$               // Model aggregation; $\lambda_k$ is relative support set size
14:         $\Gamma^r = \left\{ \sum_{k=1}^K \lambda_k \Gamma_k^r(c) | c = 1, 2, ..., N_c \right\}$  // Prototype aggregation
15:     **end for**
16:     **for** each participant $k$ **in parallel do**
17:         Download $\theta^R, \Gamma^{R-1}$ from the server.
18:         Compute local prototypes based on $Q_k$.
19:         $\theta_k^0 \leftarrow \theta^0 - \beta \nabla_{\theta^R} f(\theta^R, Q_k)$      // Local meta-update of $\theta^0$ with query set $Q_k$ and GPAL
20:     **end for**
21:     $\phi^{t+1} = \sum_{k=1}^K \mu_k \theta_k^0$          // Aggregation of meta-updated models; $\mu_k$ is relative data size
22: **end for**

---

Before training begins, each client $k$ divides its local dataset into support set $S_k$ and query set $Q_k$. To create a training environment matching the actual $R$-rounds of FL followed by inference at deployment, in each episode of our meta-training phase, we update the model over $R$ federated rounds using the support set and then makes a final adjustment (meta-update) using the query set. In other words, the support sets are utilized for learning how to solve the task, by performing $R$ rounds of FL. The query sets are used for evaluating the performance on this task and performing the meta-update process. This overall process is repeated as the model is exposed to a series of episodes.

The detailed procedure of our meta-training is given in Algorithm 1. For a quick summary, as each episode $t$ begins, the server selects a new set of $K$ participants. The model $\phi^t$, carried over from the last episodic stage, becomes the initial model $\theta^0$ for the current episode. After $R$ rounds of FL with each round consisting of local updates via local support sets and a global aggregation, $\theta^0$ evolves to $\theta^R$. Before moving to the next episode, local meta-updates are done based on $\theta^R$ using the local query sets to adjust the initial model $\theta^0$, in the spirit of MAML. As these meta-updated models get aggregated to $\phi^{t+1}$ at the server, the new episode can begin.

### 2.2.1  $R$ Rounds of Local Updates and Aggregations

In defining the loss function, we utilize the class prototypes and associated Euclidean distance metric of [18], a proven method of simple yet effective learning of embedding space. For each communication round $r$, we not only aggregate the global model $\theta^{r+1}$ but also the global prototypes $\Gamma^r = \{\Gamma^r(c) | c = 1, 2, ..., N_c\}$ for all classes, where $N_c$ is the number of classes over all clients. The class prototype is the class-specific averaged feature for data samples and calculated as Line 9 in Algorithm 1.

**Model and global prototype download (Line 7).** In the beginning of round $r \geq 1$, the server has the global model $\theta^r$ and the global prototypes $\Gamma^{r-1} = \{\Gamma^{r-1}(c) | c = 1, 2, ..., N_c\}$ from the previous round $r - 1$. Each participant $k$ first downloads $\theta^r$ and $\Gamma^{r-1}$ from the server. Since there is no global prototype in the first round, the participants only download the model $\theta^0$ when $r = 0$.

**Local prototype calculation (Line 9).** The local prototype of $\Gamma_k^r(c)$ for participant $k$ is computed as in Line 9 using the downloaded model $\theta^r$, the associated embedder outputs $g_\theta$ corresponding to the local support samples $S_k(c)$ labeled $c$. This local prototype serves as a representative of class $c$ calculated based on the local data (support set) of client $k$.

**Loss calculation from local prototypes.** Let $\Gamma_k^r$ be the set of all classes of prototypes at participant $k$: $\Gamma_k^r = \{\Gamma_k^r(c)|c \in C_k\}$, where $C_k$ is a set of all classes at participant $k$. Now using $S_k$, $\theta^r$ and $\Gamma_k^r$, each participant $k$ computes the local loss according to

$$L_{\text{local}}^{S_k}(\theta, \Gamma_k^r(c)) = \frac{1}{\sum_{c \in C_k}|S_k(c)|} \sum_{c \in C_k} \sum_{x \in S_k(c)} \left\{ d\big(g_\theta(x), \Gamma_k^r(c)\big) + \log \sum_{c' \neq c} \exp\big(-d\big(g_\theta(x), \Gamma_k^r(c')\big)\big) \right\}, \tag{2}$$

based on Euclidean distance $d(\cdot)$ between $\Gamma_k^r(c)$ and $g_\theta(x)$ for $x \in S_k(c)$.

**Auxiliary loss from global prototypes.** Relying only on the loss function of (2) based on the local prototype tends to bias the model, especially when data distributions across different clients are non-IID. This generally leads to a performance degradation of the global model. To get around, we propose a global prototype-assisted learning (GPAL) strategy, where the global prototypes serve as prior knowledge in a form of regularization to prevent local models from overfitting to their local data. Moreover, the global prototypes, reflecting classes not limited to the local dataset, can assist the local model to learn a more general embedding space. Given the global prototypes $\Gamma^{r-1} = \{\Gamma^{r-1}(c)|c = 1, 2, ..., N_c\}$ and $\{g_\theta(x)|x \in S_k\}$, the auxiliary loss $L_{\text{aux}}^{S_k}(\theta^r, \Gamma^{r-1})$ can be computed by replacing local prototypes $\Gamma_k^r$ with global prototypes $\Gamma^{r-1}$ in (2).

**Local update based on GPAL (Line 11).** Based on the local loss $L_{\text{local}}^{S_k}(\theta, \Gamma_k^r)$ computed using local prototypes and the auxiliary loss $L_{\text{aux}}^{S_k}(\theta, \Gamma^{r-1})$ based on global prototypes, the objective function becomes

$$f(\theta^r, S_k) = \gamma L_{\text{local}}^{S_k}(\theta^r, \Gamma_k^r) + (1 - \gamma)L_{\text{aux}}^{S_k}(\theta^r, \Gamma^{r-1}) \tag{3}$$

where $\gamma$ is a balancing coefficient. For $r = 0$, we have $f(\theta^r, S_k) = L_{\text{local}}^{S_k}(\theta^r, \Gamma_k^r)$ since the global prototype is not defined in the first global round. Line 11 of Algorithm 1 performs local update accordingly, where $\alpha$ is the learning rate. We call this strategy GPAL.

In FL, the clients can perform multiple local updates, say $E$ times. Hence, the process of local prototype computation in Line 9 of Algorithm 1, loss computation in (3) and local update of Line 11 can be repeated $E$ times to obtain $\theta_k^{r+1}$.

**Model and prototype aggregations (Lines 13~14).** After performing local updates, each participant $k$ sends its updated local model $\theta_k^{r+1}$ and the computed local prototypes $\Gamma_k^r$ to the server. Then the server aggregates them according to Lines 13 and 14 in Algorithm 1, where the weighting factor $\lambda_k = \frac{|S_k|}{\sum_{j=1}^K |S_j|}$ reflects the relative support set sizes.

The above local update and global aggregation processes are repeated for $R$ global rounds ($r = 0, 1, ..., R - 1$), and the server obtains $\theta^R$ and $\Gamma^{R-1}$ in a given episode.

### 2.2.2 One-Round Local Meta-Update and Aggregation (Lines 16~21)

Towards the end of each episode processing stage, the participants download $\theta^R$ and $\Gamma^{R-1}$ from the server. Each participant $k$ uses its query set $Q_k$ to compute the local prototypes $\Gamma_k^R$ as in as in Line 9. The query loss $f(\theta^R, Q_k)$ is calculated similar to (3) based on $Q_k$, $\theta^R$, $\Gamma^{R-1}$ and $\Gamma_k^R$. The meta-update would call for taking the derivative of this loss with respect to $\theta^0$: $\nabla_{\theta^0} f(\theta^R, Q_k) = \nabla_{\theta^R} f(\theta^R, Q_k) \times \frac{\partial \theta^R}{\partial \theta^0} = \nabla_{\theta^R} f(\theta^R, Q_k) \times \big(\prod_{r=0}^{R-1} \sum_{j=1}^K \lambda_j \frac{\partial}{\partial \theta^r}(\theta^r - \alpha \nabla_{\theta^r} f(\theta^r, S_j))\big)$. But one would need the double derivatives from other user locations as well, which is highly inconvenient. Ignoring the double derivative terms, we simply replace $\nabla_{\theta^0} f(\theta^R, Q_k)$ with $\nabla_{\theta^R} f(\theta^R, Q_k)$, as in Line 19. This is the same as making a first-order approximation to the MAML-like meta-update, as often done in the implementation of MAML variants including the original work of [5]. All our reported experimental results as well as convergence analysis in the present paper reflect this choice. The server finally aggregates the meta-updated models from all participants. The next episode begins as the server selects a new set of $K$ participants.

### 2.3 Testing (Deployment Stage)

In the actual deployment or test phase, given a group of clients, the server sets $\theta^0 = \phi^T$ and then leads $R$ rounds of FL to obtain $\theta^R$ and $\Gamma^{R-1}$. Now given a test sample, we make prediction based on $\theta^R$ and $\Gamma^{R-1}$: the model output is first computed using $\theta^R$ and then comparison is made with the distances from all global prototypes in $\Gamma^{R-1}$ to reach a decision.

# 3 Convergence Analysis

We provide theoretical analysis to guarantee a certain convergence behavior for our meta-training algorithm for nonconvex loss functions $f_k(\phi) := f(\phi, D_k)$. We need the following assumptions commonly made in convergence analyses of FL involving meta-learning, e.g., [12, 4].

**Assumption 1.** *For all $i$, $f_i$ is $L$-smooth, i.e., $\|\nabla f_i(\phi_1) - \nabla f_i(\phi_2)\| \leq L \|\phi_1 - \phi_2\|$ for any $\phi_1$, $\phi_2$.*

**Assumption 2.** *Let $l_i(\phi; x)$ be the loss function for a single data point $x \in D_i$ of participant $i$. For all $i = 1, 2, ..., N$, the variance of the loss gradients across data samples at a given participant is bounded, i.e., $\mathbb{E}_{x \in D_i}[\|\nabla l_i(\phi; x) - \nabla f_i(\phi)\|^2] \leq V_d$ for any $\phi$.*

**Assumption 3.** *Let $f(\phi) = \frac{1}{N} \sum_{i=1}^{N} f_i(\phi)$ be the average local loss of all participants in the system. The variance of the gradient of loss $f_i$ across participants is bounded, i.e., $\frac{1}{N} \sum_{i=1}^{N} \|\nabla f_i(\phi) - \nabla f(\phi)\|^2 \leq V_p$ for any $\phi$.*

Two key lemmas and a theorem below establish the convergence of our method. All proofs are in Supplementary Material.

**Lemma 1.** *Assume that the learning rate $\alpha$ is in the range $(0, 1/L]$. Then, the global loss function $F(\phi)$ in (1) is $L_F$-smooth, where $L_F = L2^R$.*

**Lemma 2.** *Define the local loss of our scheme $F_k(\phi) := f_k(\theta^R(\phi))$ at participant $k$. For a group $A$ with $K$ clients, define the loss averaged within that group $\mathcal{F}_A(\phi) := \frac{1}{K} \sum_{k \in A} F_k(\phi)$. Assume $\alpha \in (0, 1/L]$. Then, the variance of the gradient of $\mathcal{F}_A(\phi)$ across groups is bounded as*

$$|\mathcal{A}|^{-1} \sum_{A \in \mathcal{A}} \|\nabla \mathcal{F}_A(\phi) - \nabla F(\phi)\|^2 \leq V_p K^{-1} \tag{4}$$

*where $\mathcal{A}$ is the set of all possible groupings of $K$ participants drawn from a pool of $N$ individuals.*

**Theorem 1.** *Suppose Assumptions 1, 2, 3 hold and $\alpha \in (0, 1/L]$. Let $|D|$ be the mini-batch size at the meta-update processes of all participants. Then, Algorithm 1 guarantees the following upper bound on the loss gradient associated with our learned model $\phi^T$:*

$$\frac{1}{T} \sum_{t=0}^{T-1} \mathbb{E}[\|\nabla F(\phi^t)\|^2] \leq \frac{4(F(\phi^0) - F(\phi^\star))}{\beta T} + \epsilon(\beta, R, |D|, K) \tag{5}$$

*where $\phi^\star$ is the optimal solution of (1) and $\epsilon(\beta, R, |D|, K) = \beta L2^{R+2}(V_d |D|^{-1} + V_p K^{-1})$.*

As the number of episodes $T$ increases, the upper bound of (5) settles to $\epsilon$. For a given smoothness $L$, assumed loss gradient variance bounds $(V_d, V_p)$ and a targeted number of FL rounds $R$, the error term $\epsilon$ is controlled by the meta-update learning rate $\beta$, the mini-batch size $|D|$ and the per-episode number of participants $K$. For any reasonable value $R$, practical choices of $\beta$, $|D|$ and $K$ can make $\epsilon$ sufficiently small, as discussed in in Supplementary Material using representative parameter values.

# 4 Experiments

We validate our algorithm on CIFAR-100 [10], *mini*ImageNet [19], FEMNIST[2]. Following the data splits in [14], for CIFAR-100 and *mini*ImageNet, 100 classes are divided into 64 train, 16 validation and 20 test classes. For FEMNIST, we divide 62 classes into 52 alphabet (uppercase, lowercase) and 10 digit classes. For each class of FEMNIST, we sort the images by its name and choose first 600 samples. After all, we have 600 samples for each class in every dataset. 52 alphabet classes are set to train classes, while 10 digit classes are set to test classes. The train classes are utilized for the preparation stage, and the test classes are utilized at deployment to model the unseen tasks.

**Comparison schemes.** First, as a simplest baseline, we consider FedAvg [13], where a randomly initialized model is trained for $R$ FL rounds at deployment. The preparation stage is not considered for this scheme. Thus, direct performance comparison would be obviously unfair for FedAvg, but we just want to show what kind of performance improvement is possible by meta-learned initialization versus random initialization. Second, we consider a FedAvg-based fine-tuning, where the model is first pretrained by conducting FedAvg in each episode during preparation, and then fine-tuned with new clients for $R$ FL rounds via FedAvg at deployment. For example, in *mini*ImageNet, a 64-way classifier model is first pretrained in the preparation stage. Next, the last linear layer is replaced by a

Table 1: Performance with only unseen classes at deployment in an **IID** setup.

| Methods | CIFAR-100 | *mini*ImageNet | FEMNIST |
|---|---|---|---|
| FedAvg | $51.55 \pm 0.38\%$ | $38.80 \pm 0.26\%$ | $74.76 \pm 0.35\%$ |
| Fine-tuning via FedAvg | $63.18 \pm 0.41\%$ | $61.58 \pm 0.47\%$ | $91.95 \pm 0.28\%$ |
| Fine-tuning via one-shot FL [6] | $64.71 \pm 0.37\%$ | $65.23 \pm 0.43\%$ | $93.62 \pm 0.26\%$ |
| **FRL**: Linear classifier (Ours) | $67.32 \pm 0.37\%$ | $67.75 \pm 0.35\%$ | $94.86 \pm 0.13\%$ |
| **FRL**: Distance-based classifier (Ours) | $69.74 \pm 0.31\%$ | $68.05 \pm 0.34\%$ | $95.07 \pm 0.10\%$ |
| **FRL**: Distance-based classifier + GPAL (Ours) | $\mathbf{72.93 \pm 0.32}\%$ | $\mathbf{69.31 \pm 0.33}\%$ | $\mathbf{96.61 \pm 0.09}\%$ |

Xavier-initialized layer, and then the overall model is fine-tuned to the group at deployment. We also consider fine-tuning based on one-shot FL [6], where the local models are sampled and aggregated by ensemble cross-validation. We allow a larger number of available clients (in the deployment stage) for this scheme to accommodate user sampling. The model is first pretrained via FedAvg during preparation, and then fine-tuned based on the scheme of [6] for $R$ rounds at deployment. Finally, although comparison with personalized FL [4] is tricky as the goal is different, a global model can still be trained by repeating local updates and aggregations for $R$ FL rounds starting from the initialized model geared to client personalization. Comparison results with this "forced" global model are given in Supplementary Material. For our few-round learning (FRL), we try both a linear classifier and a distance-based classifier [18] for comparison. For the linear classifier, we connect an additional linear layer behind CNN layers, as in other baselines. The distance-based classifier utilizes prototypes instead of using the linear layer. For the distance-based classifier that utilizes prototypes, we observe the effect of our GPAL strategy. Although we utilize FedAvg for the model aggregation at the server, adopting other aggregation methods that outperform FedAvg can further improve the performance of our method and other baselines.

**Preparation stage.** We assume $N = 64$ participants in the system in the preparation stage for CIFAR-100 and *mini*ImangeNet. We assume $N = 52$ for FEMNIST. For every dataset, following [13], training data samples are prepared into $2N$ shards of 300 samples each, such that each shard corresponds to one image class. Each participant is given two shards, and these two shards may belong to either a common class or two distinct classes. This models non-IID data distributions across participants. To construct each episode, the server then randomly selects $K = 10$ out of $N$ participants. Each participant uses one half of its local data from each class as support samples, and the remaining half as query samples. We typically set the target number of global rounds to $R = 3$. Each episode of our scheme requires 4 global rounds in the meta-training phase: 3 rounds of local updates and aggregation, and 1 round of local meta-update and aggregation. For a fair comparison, we let all baselines to consume the same amount of communication resources in the preparation stage: up to $40,000$ communication rounds between the server and participants (other than FedAvg that employs no preparation rounds). Hence, our scheme is meta-trained over up to $10,000$ episodes, taking 4 rounds in each episode. We also reiterate that model preparation at the service provider is not a real-time requirement and can be done when bandwidth demands are sparse; this offers an even more favorable performance/complexity tradeoff options for the proposed scheme.

**Deployment stage.** At deployment, we distribute the initial model obtained in the preparation stage to a new group of clients. To measure the performance, we obtain the average test accuracy with a 95% confidence interval over 1000 different groups (with $K = 10$ clients in each group) after $R$ rounds of FL. For testing, in one case we randomly sample $\tau$ classes from test classes that have not been seen during preparation and distribute across $K = 10$ clients. In the other case, we randomly sample $\tau_u$ classes from the unseen test classes and $\tau - \tau_u$ classes from the train classes seen during meta-learning. We consider both IID and non-IID distributions. In the IID setup, the data samples from each class are equally distributed to $K = 10$ clients. In the non-IID setup, we distribute data as in the preparation stage. The support set is utilized for $R$ FL rounds and the server calculates test accuracy with the global model and the gathered query sets of all clients. For the one-shot FL scheme, we allow 20 clients and the server samples $K = 10$ of them to aggregate. We focus on a 5-way setup (i.e., $\tau = 5$) in the main paper with the $\tau = 10$ case reported in Supplementary Material.

**Implementation details.** The structure of the model follows the setting of [5] and [18], containing 4 consecutive $3 \times 3$ convolutional layers with 64 filters. Successively, each CNN output goes through batch normalization, ReLU, and $2 \times 2$ max pooling. In the case of CIFAR-100 where the size of image is $32 \times 32$, the last two max pooling layers are omitted to up-scale the feature map. We adopt the SGD optimizer with a learning rate of $\beta = 0.001$ for the meta-learner and a learning rate of $\alpha = 0.0001$ for

Table 2: Performance with only unseen classes at deployment in a **non-IID** setup.

| Methods | CIFAR-100 | *mini*ImageNet | FEMNIST |
|---|---|---|---|
| FedAvg | $34.85 \pm 0.27\%$ | $29.74 \pm 0.22\%$ | $59.22 \pm 0.18\%$ |
| Fine-tuning via FedAvg | $44.33 \pm 0.37\%$ | $33.39 \pm 0.41\%$ | $58.23 \pm 0.65\%$ |
| Fine-tuning via one-shot FL [6] | $35.11 \pm 0.46\%$ | $27.16 \pm 0.42\%$ | $57.88 \pm 0.67\%$ |
| **FRL**: Linear classifier (Ours) | $52.98 \pm 0.42\%$ | $53.51 \pm 0.43\%$ | $85.14 \pm 0.44\%$ |
| **FRL**: Distance-based classifier (Ours) | $63.85 \pm 0.43\%$ | $61.07 \pm 0.41\%$ | $88.60 \pm 0.42\%$ |
| **FRL**: Distance-based classifier + GPAL (Ours) | $\mathbf{66.87 \pm 0.40\%}$ | $\mathbf{63.41 \pm 0.39\%}$ | $\mathbf{92.42 \pm 0.32\%}$ |

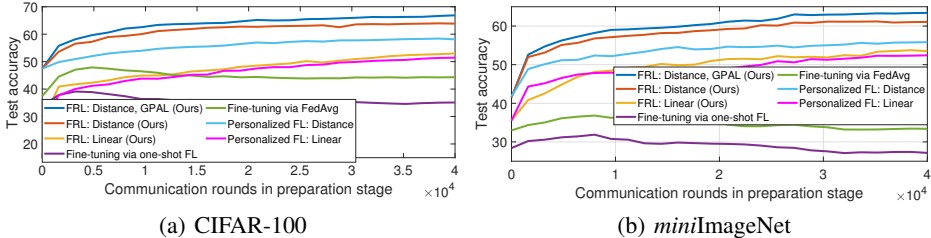

(a) CIFAR-100            (b) *mini*ImageNet

Figure 2: Final test accuracy at deployment, with varying numbers of communication rounds (proportional to the number of episodes for FRL and to the number of pretraining rounds for fine-tuning) in the preparation stage.

the learner. We set the mini-batch size to 60 and the number of local epochs at each client to $E = 1$. All methods are implemented using Pytorch and trained with a single GeForce RTX 2080 Ti.

**Results with unseen classes at deployment.** Tables 1 and 2 show test accuracies averaged over 1000 different groups after $R = 3$ global rounds at deployment, where the goal of each group is to classify $\tau = 5$ classes that were unseen during preparation. First, it can be seen that FedAvg yields significantly lower accuracy compared to others, as expected, since it uses a randomly initialized model for training. By pretraining the model, FedAvg-based fine-tuning gives significant performance gains compared to naive application of FedAvg, underlying the importance of initialization efforts. The fine-tuning scheme based on one-shot FL shows further performance improvements in the IID setup. However, since $K = 10$ clients are sampled from 20 clients for this method, there possibly exist some unseen classes when building the global model in the non-IID setup, which lowers the performance compared to fine-tuned FedAvg. Our FRL algorithm performs the best, with the distance-based classifier showing better accuracy compared to the linear classifier. The relative gains of our methods for non-IID are particularly strong. It can be also seen that the performance of the global model can be further improved by our GPAL strategy. Fig. 2 shows how the final test accuracy (after 3 fixed FL rounds at deployment) improves with the number of communication rounds in the preparation stage. The overall results in Tables 1, 2 and Fig. 2 confirm the advantage of exploiting meta-learning and global prototype-assisted learning to facilitate few-round FL.

**Results with both unseen/seen classes at deployment.** In Table 3, we report test accuracies with both unseen/seen classes at deployment; the goal of each group is to classify $\tau = 5$ classes, 2 from the unseen classes and 3 from the seen classes. Since the tasks also handle classes already seen during preparation, the accuracies are generally higher than in Tables 1, 2. The trend is consistent with the results in Tables 1 and 2, confirming the advantage of the proposed algorithm.

**Effect of global prototype-assisted learning.** To understand the effect of our GPAL method further, we visualized t-SNE of the embedding space at a client in Fig. 3. CIFAR-100 is considered with

Table 3: Performance with both unseen/seen classes at deployment.

| Methods | CIFAR-100 | | *mini*ImageNet | |
|---|---|---|---|---|
| | IID | Non-IID | IID | Non-IID |
| FedAvg | $50.03 \pm 0.42\%$ | $34.82 \pm 0.31\%$ | $42.17 \pm 0.36\%$ | $30.37 \pm 0.26\%$ |
| Fine-tuning via FedAvg | $66.73 \pm 0.36\%$ | $44.46 \pm 0.36\%$ | $63.82 \pm 0.49\%$ | $36.18 \pm 0.42\%$ |
| Fine-tuning via one-shot FL [6] | $69.84 \pm 0.39\%$ | $35.33 \pm 0.46\%$ | $67.05 \pm 0.44\%$ | $29.12 \pm 0.43\%$ |
| **FRL**: Linear classifier (Ours) | $68.22 \pm 0.38\%$ | $53.62 \pm 0.45\%$ | $69.02 \pm 0.39\%$ | $55.18 \pm 0.46\%$ |
| **FRL**: Distance-based classifier (Ours) | $70.49 \pm 0.36\%$ | $65.13 \pm 0.43\%$ | $70.39 \pm 0.38\%$ | $62.42 \pm 0.43\%$ |
| **FRL**: Distance-based classifier + GPAL (Ours) | $\mathbf{73.68 \pm 0.37\%}$ | $\mathbf{67.31 \pm 0.44\%}$ | $\mathbf{71.81 \pm 0.34\%}$ | $\mathbf{65.33 \pm 0.42\%}$ |

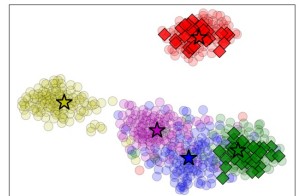 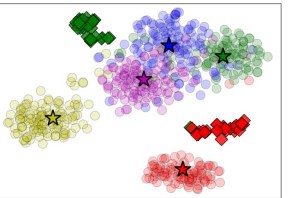 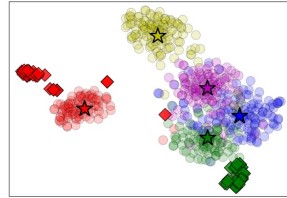

(a) Before update          (b) After update without GPAL   (c) After update with GPAL

Figure 3: t-SNE visualization of embedding space at a client. The local data samples of the client is illustrated with diamond $\diamondsuit$, and the data points of all other clients are represented by circle $\bigcirc$. The global prototypes of each class are shown with $\star$. GPAL prevents the model from being biased to its local data and enables to learn more general embedding space. This leads to performance improvements as seen in Tables 1, 2, 3 and Fig. 2.

each client having two classes in its local data in a non-IID setup. When only local prototypes are used for training as in Fig. 3(b), it can be seen that the two classes of the client form clusters without considering the data samples of other clients (but still well-separated). By considering the global prototypes (reflecting classes of all participants), in Fig. 3(c), the data points in the local client form clusters while staying away from all other global prototypes, a clearly desirable feat. This prevents the local model from being biased to its local data and enables the local model to learn a more general embedding space compared to the case in Fig. 3(b) considering only the local prototypes.

**Other experimental results.** Additional results on other settings including higher-way classification, larger group size and mismatched $R$ are reported in Supplementary Material. Comparison with the "foreced" global model based on the personalization scheme is also shown in Supplementary Material.

## 5   Conclusion

We proposed a meta-learning strategy to prepare an initial model geared to few-round federated learning. Given a group of clients with a new task, our meta-trained model generalizes well within only a few FL rounds. Convergence of our meta-training is guaranteed through theoretical analysis. Extensive experimental results confirm significant advantages of our idea over different baselines such as FedAvg-based fine-tuning and personalized FL in various setups. Our solution offers a promising direction for FL in practice, where minimizing training time and communication resources required in real-time is among key challenges.

## Acknowledgments

This work was supported by IITP fund from MSIT of Korea (No. 2020-0-00626) and by National Research Foundation of Korea (No. 2019R1I1A2A02061135).

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
