# Few-Round Learning for Federated Learning
# (Supplementary Material)

**Younghyun Park**\*
dnffkf369@kaist.ac.kr

**Dong-Jun Han**\*
djhan93@kaist.ac.kr

**Do-Yeon Kim**
dy.kim@kaist.ac.kr

**Jun Seo**
tjwns0630@kaist.ac.kr

**Jaekyun Moon**
jmoon@kaist.edu

School of Electrical Engineering,
Korea Advanced Institute of Science and Technology (KAIST)

## A    Experiments with Mismatched $R$

In Table 1, we show the results for mismatched $R$, where the actual number of rounds at deployment turns out to be smaller than expected during preparation. Specifically, the model is meta-trained targeting $R = 3$, but actual test accuracies are evaluated after $R = 2$ global rounds in the deployment stage. We randomly sample $\tau = 5$ classes from 20 test classes that have not been seen during preparation and distribute across $K = 10$ clients at deployment. Since we utilize only $R = 2$ global rounds, accuracies are generally lower than the results in the main manuscript with $R = 3$. However, the trend is consistent with the results in the main manuscript, confirming the advantages of our algorithm even in the mismatch setup.

Table 1: Mismatch scenario: The model is meta-trained targeting $R = 3$, but tested after $R = 2$ global rounds at deployment.

| Methods | CIFAR-100 | | miniImageNet | |
|---|---|---|---|---|
| | IID | Non-IID | IID | Non-IID |
| FedAvg | $44.41 \pm 0.39\%$ | $30.61 \pm 0.25\%$ | $37.31 \pm 0.29\%$ | $25.70 \pm 0.20\%$ |
| Fine-tuning via FedAvg | $59.83 \pm 0.40\%$ | $41.38 \pm 0.37\%$ | $57.40 \pm 0.59\%$ | $33.56 \pm 0.40\%$ |
| Fine-tuning via one-shot FL | $60.53 \pm 0.39\%$ | $33.62 \pm 0.41\%$ | $60.25 \pm 0.53\%$ | $27.36 \pm 0.37\%$ |
| **FRL**: Linear classifier (Ours) | $65.83 \pm 0.38\%$ | $53.17 \pm 0.40\%$ | $66.06 \pm 0.36\%$ | $51.32 \pm 0.46\%$ |
| **FRL**: Distance-based classifier (Ours) | $69.55 \pm 0.31\%$ | $63.99 \pm 0.39\%$ | $67.95 \pm 0.33\%$ | $60.56 \pm 0.41\%$ |
| **FRL**: Distance-based classifier + GPAL (Ours) | $\mathbf{72.65 \pm 0.32}\%$ | $\mathbf{66.07 \pm 0.42}\%$ | $\mathbf{69.22 \pm 0.32}\%$ | $\mathbf{63.26 \pm 0.41}\%$ |
| Personalized FL: Linear classifier | $60.41 \pm 0.32\%$ | $50.83 \pm 0.42\%$ | $61.17 \pm 0.36\%$ | $51.53 \pm 0.45\%$ |
| Personalized FL: Distance-based classifier | $61.05 \pm 0.34\%$ | $57.39 \pm 0.37\%$ | $67.02 \pm 0.35\%$ | $54.96 \pm 0.39\%$ |

## B    Experiments with a Larger Group Size

To demonstrate the scalability of the proposed method, we performed additional experiments with $K = 50$ participants; we construct each episode with $K = 50$ participants in the preparation stage, and averaged the performance of 1000 groups each having $K = 50$ clients. Other setups are exactly the same in the main manuscript. Table 2 shows the results in a 5-way setup at deployment, indicating that our scheme still outperforms other baselines in a larger scale federated learning system.

---

\*Equal contribution.

35th Conference on Neural Information Processing Systems (NeurIPS 2021).

Table 2: Experiments with group size $K = 50$: Meta-trained targeting $R = 3$ in the preparation stage, and tested after $R = 3$ global rounds in the deployment stage.

| Methods | CIFAR-100 | |
| --- | --- | --- |
| | IID | Non-IID |
| FedAvg | 27.54 ± 0.25% | 23.47 ± 0.16% |
| Fine-tuning via FedAvg | 52.16 ± 0.42% | 49.67 ± 0.33% |
| Fine-tuning via one-shot FL | 52.22 ± 0.40% | 28.47 ± 0.36% |
| **FRL**: Linear classifier (Ours) | 61.18 ± 0.40% | 53.17 ± 0.36% |
| **FRL**: Distance-based classifier (Ours) | 69.52 ± 0.32% | 69.13 ± 0.33% |
| **FRL**: Distance-based classifier + GPAL (Ours) | **72.19 ± 0.22**% | **70.11 ± 0.32**% |
| Personalized FL: Linear classifier | 60.81 ± 0.41% | 53.01 ± 0.34% |
| Personalized FL: Distance-based classifier | 63.79 ± 0.35% | 62.93 ± 0.33% |

## C  Experiments in a 10-way Setup ($\tau = 10$)

In the deployment stage of Table 3, we randomly sampled $\tau = 10$ classes (instead of 5 classes) from 20 test classes that have not been seen during preparation and distribute across $K = 10$ clients. The accuracies are lower than the 5-way setup, which is natural given the more complicated task, but the trend is still consistent with the previous results, confirming significant advantages of our meta-trained initialization.

Table 3: Experiments in a 10-way setup in the deployment stage: Meta-trained targeting $R = 3$ , and tested after $R = 3$ global rounds at deployment.

| Methods | CIFAR-100 | *mini*ImageNet |
| --- | --- | --- |
| | Non-IID | Non-IID |
| FedAvg | 12.45 ± 0.11% | 12.85 ± 0.07% |
| Fine-tuning via FedAvg | 29.87 ± 0.21% | 21.29 ± 0.17% |
| Fine-tuning via one-shot FL | 23.62 ± 0.19% | 16.94 ± 0.21% |
| **FRL**: Linear classifier (Ours) | 30.77 ± 0.21% | 33.49 ± 0.24% |
| **FRL**: Distance-based classifier (Ours) | 43.93 ± 0.25% | 35.02 ± 0.20% |
| **FRL**: Distance-based classifier + GPAL (Ours) | **46.31 ± 0.28**% | **38.17± 0.21**% |
| Personalized FL: Linear classifier | 34.32 ± 0.28% | 27.46 ± 0.65% |
| Personalized FL: Distance-based classifier | 43.54 ± 0.21% | 35.19 ± 0.21% |

## D  Experiments with Target $R = 5$

While the results shown in the main manuscript utilized the model targeting $R = 3$, in Table 4 we show the results with target $R = 5$, in the 5-way setup at deployment. The model is meta-trained targeting $R = 5$ in the preparation stage and the accuracies are evaluated after $R = 5$ global rounds at deployment. Overall results show that our few-round learning algorithm outperforms other baselines in this $R = 5$ as well, especially with non-IID data distributions.

Table 4: Experiments with target $R = 5$: Meta-trained targeting $R = 5$ in the preparation stage, and tested after $R = 5$ global rounds in the deployment stage.

| Methods | CIFAR-100 | | *mini*ImageNet | |
| --- | --- | --- | --- | --- |
| | IID | Non-IID | IID | Non-IID |
| FedAvg | 54.71 ± 0.41% | 40.11 ± 0.33% | 44.26 ± 0.33% | 35.23 ± 0.28% |
| Fine-tuning via FedAvg | 65.97 ± 0.37% | 46.75 ± 0.35% | 64.32 ± 0.45 % | 37.18 ± 0.41% |
| Fine-tuning via one-shot FL | 67.66 ± 0.34% | 39.45 ± 0.49% | 66.39 ± 0.37% | 29.15 ± 0.44% |
| **FRL**: Linear classifier (Ours) | 71.45 ± 0.32% | 55.87 ± 0.41% | 67.78 ± 0.35% | 54.85 ± 0.43% |
| **FRL**: Distance-based classifier (Ours) | 69.98 ± 0.31% | 63.74 ± 0.39% | 67.48 ± 0.35% | 60.24 ± 0.42% |
| **FRL**: Distance-based classifier + GPAL (Ours) | **72.83 ± 0.32**% | **66.34 ± 0.41**% | **70.41 ± 0.32**% | **63.61 ± 0.41**% |

Table 5: Comparison with personalized FL: Performance with only unseen classes at deployment in an **IID** setup.

| Methods | CIFAR-100 | *mini*ImageNet | FEMNIST |
|---|---|---|---|
| FedAvg | $51.55 \pm 0.38\%$ | $38.80 \pm 0.26\%$ | $74.76 \pm 0.35\%$ |
| Fine-tuning via FedAvg | $63.18 \pm 0.41\%$ | $61.58 \pm 0.47\%$ | $91.95 \pm 0.28\%$ |
| Fine-tuning via one-shot FL [3] | $64.71 \pm 0.37\%$ | $65.23 \pm 0.43\%$ | $93.62 \pm 0.26\%$ |
| **FRL**: Linear classifier (Ours) | $67.32 \pm 0.37\%$ | $67.75 \pm 0.35\%$ | $94.86 \pm 0.13\%$ |
| **FRL**: Distance-based classifier (Ours) | $69.74 \pm 0.31\%$ | $68.05 \pm 0.34\%$ | $95.07 \pm 0.10\%$ |
| **FRL**: Distance-based classifier + GPAL (Ours) | $\mathbf{72.93 \pm 0.32}\%$ | $\mathbf{69.31 \pm 0.33}\%$ | $\mathbf{96.61 \pm 0.09}\%$ |
| Personalized FL: Linear classifier [2] | $60.87 \pm 0.31\%$ | $61.88 \pm 0.32\%$ | $93.19 \pm 0.12\%$ |
| Personalized FL: Distance-based classifier | $61.88 \pm 0.32\%$ | $67.61 \pm 0.31\%$ | $94.11 \pm 0.12\%$ |

Table 6: Comparison with personalized FL: Performance with only unseen classes at deployment in a **non-IID** setup.

| Methods | CIFAR-100 | *mini*ImageNet | FEMNIST |
|---|---|---|---|
| FedAvg | $34.85 \pm 0.27\%$ | $29.74 \pm 0.22\%$ | $59.22 \pm 0.18\%$ |
| Fine-tuning via FedAvg | $44.33 \pm 0.37\%$ | $33.39 \pm 0.41\%$ | $58.23 \pm 0.65\%$ |
| Fine-tuning via one-shot FL [3] | $35.11 \pm 0.46\%$ | $27.16 \pm 0.42\%$ | $57.88 \pm 0.67\%$ |
| **FRL**: Linear classifier (Ours) | $52.98 \pm 0.42\%$ | $53.51 \pm 0.43\%$ | $85.14 \pm 0.44\%$ |
| **FRL**: Distance-based classifier (Ours) | $63.85 \pm 0.43\%$ | $61.07 \pm 0.41\%$ | $88.60 \pm 0.42\%$ |
| **FRL**: Distance-based classifier + GPAL (Ours) | $\mathbf{66.87 \pm 0.40}\%$ | $\mathbf{63.41 \pm 0.39}\%$ | $\mathbf{92.42 \pm 0.32}\%$ |
| Personalized FL: Linear classifier [2] | $51.54 \pm 0.38\%$ | $52.42 \pm 0.42\%$ | $80.59 \pm 0.51\%$ |
| Personalized FL: Distance-based classifier | $58.11 \pm 0.39\%$ | $55.83 \pm 0.35\%$ | $88.07 \pm 0.37\%$ |

# E  Comparison with Personalized FL Scheme

We note that our formulation targets creating a *global* model while the previous works on federated meta-learning [5, 1] aim at personalized *local* models. Given these different goals, in a non-IID setup, our method can generally handle broader classes of data than existing personalization approaches. There exists a possibility, however, that locally optimized models can be used to generate a single global model, in case a need arises afterwards. This is done simply by repeating the local updates and aggregations for $R$ FL rounds starting from the initialized model geared to client personalization, using given local data. While direct comparison of our scheme with personalized FL is not possible as the goals are different, we were curious about how this "forced" globalization would fare. Tables 5 and 6 show the results, where the setup is the same as in Tables 1 and 2 in the main manuscript. Interestingly, the personalization scheme is comparable to the best fine-tuning methods but lags well behind our methods. This latter observation is expected given the different design objectives. Table 7 also indicates that our proposed method outperforms personalization scheme with both unseen/seen classes at deployment.

# F  Performance without Using First-Order Approximation

Our meta-update process is based the first-order approximation in computing the model gradient with respect to the initial model. Recall that this choice was made as computing the double derivative terms would have required extra communication bandwidth as well increased computational load. Nevertheless, we were curious about the performance gap between the schemes with and without the first-order approximation. Tables 8 and 9 show the results in a non-IID setup with CIFAR-100 and *mini*ImageNet, respectively. The number of FL rounds $R$ is set to 3 in both meta-training and actual deployment. The number of participating clients is set to 10. However, since calculating and

Table 7: Comparison with personalized FL: Performance with both unseen/seen classes at deployment.

| Methods | CIFAR-100 | | *mini*ImageNet | |
|---|---|---|---|---|
| | IID | Non-IID | IID | Non-IID |
| FedAvg | $50.03 \pm 0.42\%$ | $34.82 \pm 0.31\%$ | $42.17 \pm 0.36\%$ | $30.37 \pm 0.26\%$ |
| Fine-tuning via FedAvg | $66.73 \pm 0.36\%$ | $44.46 \pm 0.36\%$ | $63.82 \pm 0.49\%$ | $36.18 \pm 0.42\%$ |
| Fine-tuning via one-shot FL [3] | $69.84 \pm 0.39\%$ | $35.33 \pm 0.46\%$ | $67.05 \pm 0.44\%$ | $29.12 \pm 0.43\%$ |
| **FRL**: Linear classifier (Ours) | $68.22 \pm 0.38\%$ | $53.62 \pm 0.45\%$ | $69.02 \pm 0.39\%$ | $55.18 \pm 0.46\%$ |
| **FRL**: Distance-based classifier (Ours) | $70.49 \pm 0.36\%$ | $65.13 \pm 0.43\%$ | $70.39 \pm 0.38\%$ | $62.42 \pm 0.43\%$ |
| **FRL**: Distance-based classifier + GPAL (Ours) | $\mathbf{73.68 \pm 0.37}\%$ | $\mathbf{67.31 \pm 0.44}\%$ | $\mathbf{71.81 \pm 0.34}\%$ | $\mathbf{65.33 \pm 0.42}\%$ |
| Personalized FL: Linear classifier [2] | $65.09 \pm 0.32\%$ | $52.08 \pm 0.44\%$ | $62.05 \pm 0.38\%$ | $53.53 \pm 0.49\%$ |
| Personalized FL: Distance-based classifier | $68.70 \pm 0.34\%$ | $57.62 \pm 0.41\%$ | $63.63 \pm 0.35\%$ | $58.08 \pm 0.41\%$ |

storing Jacobian matrices consume a very large memory, the number of data samples in each episode is downsized during the meta-training stage. Specifically, we decrease the number of data in each episode from 6000 to 1200 in CIFAR-100, so that each user holds only 120 images. Similarly, in *mini*ImageNet, the number of data in each episode is decreased to 320 considering the size of images. Now that each client possesses less data, we decreased the batch size to 60 for CIFAR-100 and 16 for *mini*ImageNet, respectively. In the deployment stage, we do not reduce data since double derivatives are not used there. We also let $\tau = 5$ at deployment. It can be seen that the performances do improve by considering the double derivatives from other users. We stress, however, that our scheme with only fist-order derivatives still outperforms other baselines, as shown throughout the main manuscript and Supplementary Material.

Table 8: CIFAR-100: Performance of FRL with and without first-order approximation.

| | FRL with first-order approximation | FRL without first-order approximation |
|---|---|---|
| **FRL**: Linear classifier (Ours) | $36.17 \pm 0.37\%$ | $38.03 \pm 0.33\%$ |
| **FRL**: Distance-based classifier (Ours) | $58.12 \pm 0.40\%$ | $61.89 \pm 0.41\%$ |
| **FRL**: Distance-based classifier + GPAL (Ours) | $\mathbf{60.16 \pm 0.38}\%$ | $\mathbf{62.18 \pm 0.41}\%$ |

Table 9: *mini*ImageNet: Performance of FRL with and without first-order approximation.

| | FRL with first-order approximation | FRL without first-order approximation |
|---|---|---|
| **FRL**: Linear classifier (Ours) | $39.45 \pm 0.31\%$ | $42.06 \pm 0.33\%$ |
| **FRL**: Distance-based classifier (Ours) | $54.78 \pm 0.37\%$ | $56.82 \pm 0.42\%$ |
| **FRL**: Distance-based classifier + GPAL (Ours) | $\mathbf{56.41 \pm 0.38}\%$ | $\mathbf{58.85 \pm 0.41}\%$ |

# G  Proof of Lemmas

## G.1  Proof of Lemma 1

For simplicity, $\theta^r(\phi_1) =: \theta_1^r$ and $\theta^r(\phi_2) =: \theta_2^r$ for $r = 0, \ldots, R$. Note that $\theta_1^0 = \phi_1$ and $\theta_2^0 = \phi_2$. By using first-order approximation, we can write

$$\left\| \nabla_{\phi_1} F_k(\phi_1) - \nabla_{\phi_2} F_k(\phi_2) \right\| \tag{1}$$

$$= \left\| \nabla_{\theta_1^R} f_k(\theta_1^R) - \nabla_{\theta_2^R} f_k(\theta_2^R) \right\| \tag{2}$$

$$\leq L \left\| \theta_1^R - \theta_2^R \right\| \tag{3}$$

$$= L \left\| \frac{1}{K} \sum_{k \in A_t} \left( \theta_1^{R-1} - \alpha \nabla_{\theta_1^{R-1}} f_k(\theta_1^{R-1}) \right) - \frac{1}{K} \sum_{k \in A_t} \left( \theta_2^{R-1} - \alpha \nabla_{\theta_2^{R-1}} f_k(\theta_2^{R-1}) \right) \right\| \tag{4}$$

$$\leq L \frac{1}{K} \sum_{k \in A_t} \left\| \theta_1^{R-1} - \theta_2^{R-1} - \left( \alpha \nabla_{\theta_1^{R-1}} f_k(\theta_1^{R-1}) - \alpha \nabla_{\theta_2^{R-1}} f_k(\theta_2^{R-1}) \right) \right\| \tag{5}$$

$$\leq L \frac{1}{K} \sum_{k \in A_t} \left( \left\| \theta_1^{R-1} - \theta_2^{R-1} \right\| + \alpha \left\| \nabla_{\theta_1^{R-1}} f_k(\theta_1^{R-1}) - \nabla_{\theta_2^{R-1}} f_k(\theta_2^{R-1}) \right\| \right) \tag{6}$$

$$\leq L(1 + \alpha L) \left\| \theta_1^{R-1} - \theta_2^{R-1} \right\| \tag{7}$$

$$\leq L(1 + \alpha L) \frac{1}{K} \sum_{k \in A_t} \left( \left\| \theta_1^{R-2} - \theta_2^{R-2} \right\| + \alpha \left\| \nabla_{\theta_1^{R-2}} f_k(\theta_1^{R-2}) - \nabla_{\theta_2^{R-2}} f_k(\theta_2^{R-2}) \right\| \right) \tag{8}$$

$$\vdots \tag{9}$$

$$\leq L(1 + \alpha L)^R \left\| \phi_1 - \phi_2 \right\| \tag{10}$$

$$\leq L 2^R \left\| \phi_1 - \phi_2 \right\| \tag{11}$$

$$= L_F \left\| \phi_1 - \phi_2 \right\| \tag{12}$$

Hence, $F_k(\phi)$ is $L_F$-smooth where

$$L_F = L 2^R. \tag{13}$$

Now from

$$\nabla F(\phi) = \mathbb{E}_{A_t \sim p(\mathcal{A})} \left[ \frac{1}{K} \sum_{k \in A_t} \nabla F_k(\phi) \right], \tag{14}$$

we have

$$\|\nabla F(\phi_1) - \nabla F(\phi_2)\| \le \sum_{A_t \in \mathcal{A}} p(A_t) \left\| \frac{1}{K} \sum_{k \in A_t} (\nabla F_k(\phi_1) - F_k(\phi_2)) \right\| \tag{15}$$

$$\le \sum_{A_t \in \mathcal{A}} p(A_t) \left( \frac{1}{K} \sum_{k \in A_t} \|\nabla F_k(\phi_1) - F_k(\phi_2)\| \right) \tag{16}$$

$$\le \sum_{A_t \in \mathcal{A}} p(A_t) \left( \frac{1}{K} \sum_{k \in A_t} L_F \|\phi_1 - \phi_2\| \right) \tag{17}$$

$$= L_F \|\phi_1 - \phi_2\|. \tag{18}$$

Hence, $F(\phi)$ is also $L_F$-smooth, which completes the proof.

### G.2 Proof of Lemma 2

Note that

$$\nabla F(\phi) = \mathbb{E}_{A \sim p(\mathcal{A})} \left[ \frac{1}{K} \sum_{k \in A} \nabla F_k(\phi) \right] \tag{19}$$

$$= \sum_{A \in \mathcal{A}} \frac{1}{|\mathcal{A}|} \left( \frac{1}{K} \sum_{k \in A} \nabla F_k(\phi) \right). \tag{20}$$

By letting $\sigma^2 = \frac{1}{N} \sum_{k=1}^N \left\| b_k - \frac{1}{N} \sum_{k=1}^N b_k \right\|^2$, we can write

$$\frac{1}{|A|} \sum_{A \in A} \left\| \left( \frac{1}{K} \sum_{k \in A} \nabla F_k(\phi) \right) - \nabla F(\phi) \right\|^2 \tag{21}$$

$$\le \frac{1}{|\mathcal{A}|} \sum_{A \in \mathcal{A}} \left\| \left( \frac{1}{K} \sum_{k \in A} \nabla F_k(\phi) \right) - \frac{1}{|\mathcal{A}|} \sum_{A \in \mathcal{A}} \left( \frac{1}{K} \sum_{k \in A} \nabla F_k(\phi) \right) \right\|^2 \tag{22}$$

$$\underset{(a)}{\le} \frac{1}{|\mathcal{A}|} \sum_{A \in \mathcal{A}} \left\| \left( \frac{1}{K} \sum_{k \in A} \nabla_{\theta^R} f_k(\theta^R) \right) - \frac{1}{|\mathcal{A}|} \sum_{A \in \mathcal{A}} \left( \frac{1}{K} \sum_{k \in A} \nabla_{\theta^R} f_k(\theta^R) \right) \right\|^2 \tag{23}$$

$$\underset{(b)}{\le} \frac{1}{|\mathcal{A}|} \sum_{A \in \mathcal{A}} \left\| \left( \frac{1}{K} \sum_{k \in A} b_k \right) - \frac{1}{|A|} \sum_{A \in \mathcal{A}} \left( \frac{1}{K} \sum_{k \in} b_k \right) \right\|^2 \tag{24}$$

$$\underset{(c)}{\le} \frac{1}{|\mathcal{A}|} \sum_{A \in \mathcal{A}} \left\| \left( \frac{1}{K} \sum_{k \in A} b_k \right) - \frac{1}{N} \sum_{k=1}^N b_k \right\|^2 \tag{25}$$

$$\underset{(d)}{=} \frac{\sigma^2 (1 - \frac{K}{N})}{\frac{K}{N}(N-1)} \tag{26}$$

$$\underset{(e)}{\le} \frac{V_p (1 - \frac{K}{N})}{\frac{K}{N}(N-1)} \tag{27}$$

$$\le \frac{V_p}{K} \tag{28}$$

where $(a)$ comes from first-order approximation, $(b)$ comes by setting $b_k = \nabla_\phi f_k(\phi)$, $(c)$ is trivial. Note that $(d)$ comes from the fact that $\mathbb{E}_A \left[ \left\| \frac{1}{K} \sum_{k \in A} b_k - \mu \right\| \right] \le \frac{\sigma^2 (1 - \frac{K}{N})}{\frac{K}{N}(N-1)}$ with $\mu = \frac{1}{N} \sum_{k=1}^N b_k$ and $\sigma^2 = \frac{1}{N} \sum_{k=1}^N \|b_k - \mu\|^2$ [4]. See proof of Lemma 5 in [4] for the details. Finally, $(e)$ comes from Assumption 3.

## H  Proof of Theorem 1

### Step 1: Formulation

Since $F(\phi)$ is $L_F$ smooth by Lemma 1, we have

$$F(\phi^{t+1}) \leq F(\phi^t) + \nabla F(\phi^t)^T (\phi^{t+1} - \phi^t) + \frac{L_F}{2} \|\phi^{t+1} - \phi^t\|^2 \tag{29}$$

$$\leq F(\phi^t) - \beta \nabla F(\phi^t)^T \Big(\frac{1}{K} \sum_{k \in A_t} \tilde{\nabla} F_k(\phi^t)\Big) + \frac{L_F}{2} \beta^2 \|\frac{1}{K} \sum_{k \in A_t} \tilde{\nabla} F_k(\phi^t)\|^2 \tag{30}$$

where the last inequality holds since

$$\phi^{t+1} = \frac{1}{K} \sum_{k \in A_t} \Big(\phi^t - \beta \tilde{\nabla} F_k(\phi^t)\Big) \tag{31}$$

$$= \phi^t - \beta \frac{1}{K} \sum_{k \in A_t} \tilde{\nabla} F_k(\phi^t) \tag{32}$$

By taking the expectation at (30), we have

$$\mathbb{E}[F(\phi^{t+1})] \leq \mathbb{E}[F(\phi^t)] - \beta \mathbb{E}\Big[\nabla F(\phi^t)^T \Big(\frac{1}{K} \sum_{k \in A_t} \tilde{\nabla} F_k(\phi^t)\Big)\Big] + \frac{L_F}{2} \beta^2 \mathbb{E}\Big[\|\frac{1}{K} \sum_{k \in A_t} \tilde{\nabla} F_k(\phi^t)\|^2\Big]. \tag{33}$$

**Step 2: Bounding** $\mathbb{E}\Big[\nabla F(\phi^t)^T \Big(\frac{1}{K} \sum_{k \in A_t} \tilde{\nabla} F_k(\phi^t)\Big)\Big]$

By defining $X$ as

$$X = \frac{1}{K} \sum_{k \in A_t} \Big(\tilde{\nabla} F_k(\phi^t) - \nabla F_k(\phi^t)\Big), \tag{34}$$

we can write

$$\mathbb{E}\Big[\nabla F(\phi^t)^T \Big(\frac{1}{K} \sum_{k \in A_t} \tilde{\nabla} F_k(\phi^t)\Big)\Big] = \mathbb{E}\Big[\nabla F(\phi^t)^T \Big(X + \frac{1}{K} \sum_{k \in A_t} \nabla F_k(\phi^t)\Big)\Big] \tag{35}$$

$$\geq \underbrace{\mathbb{E}\Big[\nabla F(\phi^t)^T \Big(\frac{1}{K} \sum_{k \in A_t} \nabla F_k(\phi^t)\Big)\Big]}_{Z_1} - \underbrace{\|\mathbb{E}[\nabla F(\phi^t)^T X]\|}_{Z_2} \tag{36}$$

By the definition of our global loss function $F(\phi)$ and the law of total expectation, we have

$$Z_1 = \mathbb{E}[\|\nabla F(\phi^t)\|^2]. \tag{37}$$

Now we consider $Z_2$. Recall that

$$\nabla F_k(\phi) = \nabla_{\theta^R} f_k\big(\theta^R\big) \tag{38}$$

using first-order approximation. We define the stochastic gradient $\tilde{\nabla} F_k(\phi)$ as follows

$$\tilde{\nabla} F_k(\phi) = \tilde{\nabla}_{\theta^R} f_k\big(\theta^R, \tilde{D}_k\big). \tag{39}$$

Since $\tilde{\nabla}_{\theta^R} f_k\big(\theta^R, \tilde{D}_k\big)$ is an unbiased estimator of $\nabla_{\theta^R} f_k\big(\theta^R\big)$,

$$\|\mathbb{E}[\tilde{\nabla} F_k(\phi) - \nabla F_k(\phi)]\| = 0. \tag{40}$$

Moreover, since $\mathbb{E}[U^T V] \leq \frac{1}{4} \mathbb{E}[\|U\|^2] + \mathbb{E}[\|V\|^2]$ holds for any vectors $U$ and $V$, we can write

$$Z_2 = \big\|\mathbb{E}[\nabla F(\phi^t)^T X]\big\| \tag{41}$$

$$= \big\|\mathbb{E}\big[\mathbb{E}\big[\nabla F(\phi^t)^T X | \Omega\big]\big]\big\| \tag{42}$$

$$= \big\|\mathbb{E}\big[\nabla F(\phi^t)^T \mathbb{E}[X | \Omega]\big]\big\| \tag{43}$$

$$\leq \frac{1}{4} \mathbb{E}[\|\nabla F(\phi^t)\|^2] + \mathbb{E}\big[\|\mathbb{E}[X | \Omega]\|^2\big] \tag{44}$$

$$= \frac{1}{4} \mathbb{E}[\|\nabla F(\phi^t)\|^2]. \tag{45}$$

where the last equality holds from (40).

By inserting (37) and (45) to (36), we obtain

$$\mathbb{E}\Big[\nabla F(\phi^t)^T\Big(\frac{1}{K}\sum_{k\in A_t}\tilde{\nabla}F_k(\phi^t)\Big)\Big] \geq \frac{3}{4}\mathbb{E}[\|\nabla F(\phi^t)\|^2]. \tag{46}$$

**Step 3: Bounding** $\mathbb{E}\Big[\|\frac{1}{K}\sum_{k\in A_t}\tilde{\nabla}F_k(\phi^t)\|^2\Big]$

Since $\frac{1}{K}\sum_{k\in A_t}\tilde{\nabla}F_k(\phi^t) = \frac{1}{K}\sum_{k\in A_t}\nabla F_k(\phi^t) - X$, we can write

$$\|\frac{1}{K}\sum_{k\in A_t}\tilde{\nabla}F_k(\phi^t)\|^2 \leq 2\|\frac{1}{K}\sum_{k\in A_t}\nabla F_k(\phi^t)\|^2 + 2\|X\|^2 \tag{47}$$

$$= 2\|\frac{1}{K}\sum_{k\in A_t}\nabla F_k(\phi^t)\|^2 + 2\|\frac{1}{K}\sum_{k\in A_t}\Big(\tilde{\nabla}F_k(\phi^t) - \nabla F_k(\phi^t)\Big)\|^2 \tag{48}$$

$$\leq 2\|\frac{1}{K}\sum_{k\in A_t}\nabla F_k(\phi^t)\|^2 + 2\frac{1}{K}\sum_{k\in A_t}\|\tilde{\nabla}F_k(\phi^t) - \nabla F_k(\phi^t)\|^2 \tag{49}$$

where the first inequality comes from the fact that $\|A\|^2 \leq 2\|B\|^2 + 2\|C\|^2$ for $A = B + C$, and the last inequality comes from the Cauchy-Schwarz inequality. Now taking expectation at both sides, we have

$$\mathbb{E}[\|\frac{1}{K}\sum_{k\in A_t}\tilde{\nabla}F_k(\phi^t)\|^2] \leq 2\mathbb{E}[\|\frac{1}{K}\sum_{k\in A_t}\nabla F_k(\phi^t)\|^2] + 2\frac{1}{K}\sum_{k\in A_t}\mathbb{E}[\|\tilde{\nabla}F_k(\phi^t) - \nabla F_k(\phi^t)\|^2] \tag{50}$$

$$\underset{(a)}{=} 2\mathbb{E}[\|\frac{1}{K}\sum_{k\in A_t}\nabla F_k(\phi^t)\|^2] + 2\frac{1}{K}\sum_{k\in A_t}\mathbb{E}\Big[\big\|\tilde{\nabla}_{\theta^R}f_k\big(\theta^R,\tilde{D}_k\big) - \nabla_{\theta^R}f_k\big(\theta^R\big)\big\|^2\Big] \tag{51}$$

$$\underset{(b)}{\leq} 2\mathbb{E}[\|\frac{1}{K}\sum_{k\in A_t}\nabla F_k(\phi^t)\|^2] + \frac{2V_d}{|D|} \tag{52}$$

$$\underset{(c)}{=} 2\mathbb{E}[\|\nabla\mathcal{F}_{A_t}(\phi^t)\|^2] + \frac{2V_d}{|D|} \tag{53}$$

$$\underset{(d)}{=} 2\mathbb{E}[\|\nabla F(\phi^t)\|^2] + \frac{2V_p}{K} + \frac{2V_d}{|D|} \tag{54}$$

where $(a)$ comes from the first-order approximation, $(b)$ comes from Assumption 2, $(c)$ comes from the definition, (d) comes from Lemma 2.

**Step 4: Inserting the results of Steps 2 and 3**

Now by inserting (46) and (54) to (33), we obtain

$$\mathbb{E}[F(\phi^{t+1})] \leq \mathbb{E}[F(\phi^t)] - \beta(\frac{3}{4} - \beta L_F)\mathbb{E}[\|\nabla F(\phi^t)\|^2] + \beta^2 L_F\left(\frac{V_d}{|D|} + \frac{V_p}{K}\right) \tag{55}$$

$$\leq \mathbb{E}[F(\phi^t)] - \frac{\beta}{4}\mathbb{E}[\|\nabla F(\phi^t)\|^2] + \beta^2 L_F\left(\frac{V_d}{|D|} + \frac{V_p}{K}\right) \tag{56}$$

where the last inequality comes by setting $\beta L_F \leq \frac{1}{2}$ .

**Step 5: Final stage**

Summing up for all episodes $t = 0, 1, ...T - 1$, we have

$$\mathbb{E}[F(\phi^T)] \leq \mathbb{E}[F(\phi^0)] - \frac{\beta T}{4}\Big(\frac{1}{T}\sum_{t=0}^{T-1}\mathbb{E}[\|\nabla F(\phi^t)\|^2]\Big) + T\beta^2 L_F\left(\frac{V_d}{|D|} + \frac{V_p}{K}\right). \tag{57}$$

Finally from $F(\phi^*) \leq \mathbb{E}[F(\phi^T)]$, we can write

$$\frac{1}{T}\sum_{t=0}^{T-1}\mathbb{E}[\|\nabla F(\phi^t)\|^2] \leq \frac{4(F(\phi^0) - F(\phi^*))}{\beta T} + 4\beta L_F\left(\frac{V_d}{|D|} + \frac{V_p}{K}\right) \tag{58}$$

$$= \frac{4(F(\phi^0) - F(\phi^*))}{\beta T} + \beta L 2^{R+2}\left(\frac{V_d}{|D|} + \frac{V_p}{K}\right) \tag{59}$$

which completes the proof.

# I  Discussion on the Convergence Bound

Let us focus on $\epsilon$ in (59)

$$\epsilon(\beta, R, |D|, K) = \beta L 2^{R+2}\left(\frac{V_d}{|D|} + \frac{V_p}{K}\right). \tag{60}$$

For a given smoothness $L$, assumed loss gradient variance bounds ($V_d$, $V_p$) and a targeted number of FL rounds $R$, $\epsilon$ is controlled by the meta-update learning rate $\beta$, the mini-batch size $|D|$ and the per-episode number of participants $K$. Note that if $V_p = 0$, i.e., no variations in the loss gradients of different participants, then we have $\epsilon = \mathcal{O}(1)\beta L_F \frac{V_d}{|D|}$ for $L_F = L2^R$. This is equal to the bound on stochastic gradient descent for a nonconvex function that is $L_F$-smooth, and we can make $\epsilon$ close to zero by controlling $\beta$ and/or $|D|$. For example, let us say $L$ is in the range $1 \sim 10$ and $2^R$ in the range of 8 to 1024, which are reasonable. It is easy to imagine the minibatch size $|D|$ to be large enough compared to $L2^R$. Now, with a typical value of $\beta$ like 0.001, we can see that $\epsilon$ would be a tiny fraction of $V_d$.

Now consider the case with $V_p > 0$. Setting $|D|$ large enough would make $\frac{V_d}{|D|}$ small, in which case, $\epsilon$ would be dominated by $\beta L 2^{R+2}\frac{V_p}{K}$. Given some $R$, let us say that the group size $K$ is chosen to be larger or comparable to $L2^{R+2}$, which reflects reasonable practical scenarios. Finally, given the typically very small values of the learning rate $\beta$, it is easy to imagine $\epsilon$ settling to a very small fraction of $V_p$.