# OpenReview forum: "Few-Round Learning for Federated Learning"
_NeurIPS.cc/2021/Conference — NeurIPS 2021 Poster_

### Official Review · Reviewer_7HAh · 2021-07-14

**Rating:** 7
**Confidence:** 4

**Summary:**

The paper designs an initial global model which adapts to few round federated learning for possibly unseen tasks.  A meta-learning based approach with a global prototype-assisted learning strategy is built to tackle this problem.

**Limitations And Societal Impact:**

Yes

**Main Review:**

Although the authors take advantage of known techniques from MAML and prototype assisted learning strategies, I would say that this paper would be of interest for the federated learning community as the studied problem is really practical. It can be seen as a way to reduce the high communication cost introduced by the classic federated learning if every time the algorithm needs to be rerun for new tasks. The authors initiate the study of this problem and propose an efficient method with theoretical guarantees.

The paper is written quite well and I like the precise description step by step for their algorithm. However in the paragraph of “key idea” of Introduction, lines 51-53 are not really clear for who starts reading from the beginning. The notations are introduced in a sudden without precise explication such as D_k (but I got the idea of this part after reading the algorithm). This is just a minor suggestion for the flow of the paper. The experiments are conducted for image classification tasks, and the proposed method is compared with multiple baselines which are quite fair. The description of the details of the experiments are clear and well organized. The performances of the proposed method are quite convincing.

Just have a few questions related to the communication cost of the novel method:
1. For GPAL, at the deployment stage, the clients will need to train $\Gamma^{R-1}$ as well (e.g. the client needs to send the local prototypes). Will this add too much communication budget to few-round federated learning when the number of the classes is large? If the number of clients in the deployment stage is large, this may worsen the communication bottleneck.
2. To save the communication cost in total for deployment (wrt the number of rounds to reach a given accuracy), which value of $R$ is more interesting? Will there be an optimal one? I saw the supplementary material part of the dismatch R part and Target R=5, which somehow suggests that R=3 may be better. I would like to have the authors' insight on it.

In total, I would recommend this paper to be accepted.


**Time Spent Reviewing:**

6

---

> ### Author Response · Authors · 2021-08-09
> **Response to Reviewer 7HAh**
>
> We appreciate the reviewer for the review and the thoughtful comments. Below, we reply to the comments raised by the reviewer.
>
> 0. **Notations:** The notations in lines 51-53 are described in lines 24-27 of the current manuscript. We will reiterate some important notations and make explication clearer.
>
> 1. **Communication burden for sending prototypes:** We appreciate the reviewer for this comment. The size of the prototype is significantly smaller than the model size; the communication cost required for sending a single prototype is 0.057% of that required for the model. Even when the number of classes are very large, this would not cause significant burden for downloading the global prototypes, since the server having a large computing power can broadcast the global prototypes to the clients. In case of uploading the local prototypes from the clients to the server, the additional communication cost is almost negligible since each client has only a few classes in its local (i.e., biased datasets) in FL scenarios. This small additional communication cost can improve the model performance, as can be seen in Tables 1, 2, 3 of our main manuscript.
>
> 2. **Choosing an appropriate $R$ value:** Thank you for the interesting question. First, a model designed with a larger $R$ tends to have better performance, if $R$ rounds of FL can be performed in the deployment stage. However, when the actual number of rounds at deployment turns out to be smaller than expected during preparation (i.e., mismatch scenario), a model designed with a smaller $R$ can perform better. Hence, preparing the model targeting a too large $R$ can be inefficient considering the case of mismatch in the deployment stage.
>
> Again, thank you for your valuable comments. Please do let us know if you have any remaining questions/comments.
>
> Best, Authors

---

### Official Review · Reviewer_wVqa · 2021-07-14

**Rating:** 7
**Confidence:** 3

**Summary:**

This paper presents a method for using meta-learning to train an initial model, which is intended to achieve optimal performance on an arbitrary downstream tasks when trained for R rounds of federated learning (for some small R). The paper presents aa meta-learning-flavored algorithm for conducting the pretraining step, and provides theoretical bounds on the gradient of the resulting loss (Theorem 1). The paper includes some empirical results on CIFAR100, miniImageNet, and FEMNIST, with comparison to some other baselines (although no baselines exist from other works for this particular task, according to the authors).

**Limitations And Societal Impact:**

Yes.

**Main Review:**

Overall: In general, I found the paper to address a task which seems relevant to the quickly-growing area of federated learning. This seems to be an extremely relevant task to introduce to the field, and the paper's approach is logically presented. I found the theoretical analysis somewhat lacking, since Theorem 1 only bounds the loss gradients (and not the loss), which does not seem to translate into a useful measure of model accuracy (only convergence to some value of \epsilon). It is difficult to evaluate the empirical results given the baselines which are not really baselines (since they are optimized for entirely different scenarios), but the empirical results seem good. In general, I think that the paper would be a candidate for acceptance if the feedback below is satisfactorily addressed.

Major Comments:

* The authors note that no prior work has addressed this task. However, this makes it difficult to interpret the results of the empirical experiments, which essentially compare to three techniques not applicable to the "few-round" task, plus three versions of the proposed algorithm; effectively the other "Methods" (three versions of FRL) amount to an ablation study. I would encourage the authors to explore better baselines -- perhaps something from the meta-learning literature, at the very least, to give a sense for how those approaches may compare. This would give a dual "FL only" and "meta-learning only" perspective on why the combined FL+meta-learning approach is superior.

* The "non-IID" scenario described at L295 is a very weak model of non-IID-ness, so much so that I hesitate to refer to it as such. I would recommend to either find a better non-IID scenario (examples abound in the FL literature), clarify why this is comparable to other non-IID FL tasks, or remove the "non-IID" framing.

* It would be useful to provide some motivation for why meta-learning in particular is motivated in this scenario, as opposed to some form of unsupervised pretraining which has performed quite well for a large variety of downstream tasks e.g. in the NLP literature (for example, the recent work on T5).

* The authors make significant efforts to differentiate their task of so-called "few round learning" from existing FL tasks such as personalized learning. The difference is clear; however, the authors could do more to motivate the significance of this task. While I think there are clear arguments for this, it would be useful to make them explicit in the paper.

* I don't think the term "few-round learning" in the paper, nor the title itself, accurately represent the work described in this paper; the existing characterization make it sound like "doing FL in the fewest rounds possible". I do not have a great suggestion, but something more like "Meta-Learning for Few-Round Downstream FL", while less elegant, seems more accurate.

* The paper as-is is not self-contained; it relies on class prototypes and distance metrics only described in [18]. Please clearly describe the necessary components in the paper directly.

Minor Comments:

* The confusing phrase "the task of the group conducting FL" is used in both abstract and intro (L38). I would recommend something like "the downstream task for which the pretrained model will be used" instead.

* Should the equation f(...) on L52 also depend on the task, or somehow indicate that this is over all downstream tasks?

* The terms "query set" and "support set" are never formally defined in the paper, nor are the methods for assembling them clearly described, to my reading. Similarly, I do not think that "class prototypes" are clearly described. Please clarify.

* I thinkSection 2.2.1 could be better titled and organized. It is essentially a walk-through of Algorithm 1 (which could be better reflected in the title); also, each bolded section should refer to specific relevant lines in the Algorithm.

* The assumptions given in Section 3 are quite strict. Are these met in the experiments, or can we approximately compute them? It seems these could also create opportunities for an adversary to break the guarantees by injecting samples/clients which deliberately create large variance or large gradient variance. This could be worth at least a brief discussion or clarification.

* It isn't clear to me how Fig 3(c) has more "clearly desirable" (L354) properties than 3(b). Please clarify.

Typos etc.

* Fig 1 caption: "hope to classify an arbitrary task" --> hope to perform an arbitrary classification task
* What is meant by "actual inference" in Fig. 1 caption? Please clarify.
* L74: "via episodic" --> via an episodic
* L125: "diseases that would" -> diseases than would
* Line 11 of Algorithm 1 should refer to Equation (3).

**Update**: I have read the author response. This largely addresses my main concerns with the paper, enough so that I will increase my rating from 6 to 7. I do hope the authors will consider further clarification regarding the "downstream" target of the FL in the title and/or abstract; it is my opinion that readers will benefit from this clarification.

**Time Spent Reviewing:**

2hrs

---

> ### Author Response · Authors · 2021-08-09
> **Response to Reviewer wVqa**
>
> We appreciate the reviewer for the feedback and insightful comments. We will give responses to the comments raised by the reviewer.
>
> 0. **Bounding the loss gradient:** We first clarify that (as we stated in Line 260 of the main manuscript) $\epsilon$ vanishes or the loss gradient shrinks to zero. This indicates that our algorithm converges to the optimal solution of (1), guaranteeing the convergence of the proposed method. We note that analyzing this gradient of the loss function is the most common approach for guaranteeing convergence.
>
> **Major comments**
>
> 1. **Comparison with meta-learning based schemes:** We appreciate the reviewer’s point here; unfortunately, however, the only other FL scheme that has a meta-learning flavor is personalized FL. But personalized FL serves the different purpose (one of individual optimization versus group optimization), making direct comparison with ours quite awkward. Nevertheless, we devised a “forced” global model from personalized FL and made a performance comparison with ours. As can be seen in Tables 1, 2, 3 of the main manuscript, our scheme performs better, although this comparison may not be fair as we state on Lines 283-285.
>
> 2. **Comment regarding our non-IID scenario:** We stress that the non-IID scenario that we considered (which is also adopted in [McMahan17], [Sattler20]) is a strong model of non-IIDness; each client has data samples of only one or two classes (among 64 classes for CIFAR-100 and miniImageNet), making the local data of each client significantly biased and leading to a strong non-IID scenario. As the number of classes in the local dataset increases, we have a weaker non-IID scenario; in an IID setup, each client has all 64 classes in its local data.
>
> [McMahan17] H. B. McMahan et al., “Communication-efficient learning of deep networks from decentralized data,” AISTATS 2017.
>
> [Sattler20] F. Sattler et al., "Robust and communication-efficient federated learning from non-iid data," IEEE Transactions on Neural Networks and Learning Systems, 2020.
>
> 3. **Motivation of using meta-learning rather than unsupervised pretraining:** The reviewer raises an important point. We believe it is possible to apply unsupervised pretraining to get strong results given ample data and time. For the present work, however, our motivation was to tailor the training process to specific “R” rounds of federated learning and for that, meta-learning was a natural choice. Keep in mind that two of the baseline methods we compared, fine-tuning via FedAvg and fine-tuning via one-shot FL, are based on pretraining, as opposed to meta-learning. While these methods are not unsupervised pretraining on massive data, they reflect the same philosophical direction of pretraining followed by fine-tuning.
>
> 4. **Motivating the significance of our task further:** We feel creating a global model that can quickly adapt to the need of any group is significant. To us, a good motivating example is the diagnosis of a broader classes of diseases that would be possible through collaborative training across more examples contributed by a larger group of individuals (as described in Lines 123-127).
>
> 5. **Regarding the term “few-round learning”:** We like the expression “meta-learning for few-round downstream FL” that the reviewer suggested. The word “downstream” makes the intended goal clear. We appreciate the suggestion and we will make use of this expression throughout the manuscript. For the simplicity and the purposeful parallelism with “few-shot” learning, though, we would also like to maintain the original term “few-round” learning.
>
> 6. **Describing more necessary components in the paper:** Class prototype is the mean of the model output for the data samples in a specific class. We will describe the definition of class prototype more clearly. Regarding the distance metric, it is nothing but the Euclidean distance as described in Line 192 of the main manuscript; we will make this clearer as well.
>
> **Minor comments**
>
> 1. **Correcting the confusing phrase:** This is a good suggestion and we will take it.
>
> 2. **Equation $f(\cdot,\cdot)$ in line 52:** $f(\cdot,\cdot)$ in line 52 depends on the task. More precisely, the collection of local dataset of each client (that participate in FL), i.e., the collection of $D_k$, can be viewed as the downstream task that the group wants to solve in the deployment stage.
>
> 3. **Defining the support/query sets and class prototypes:** Each client randomly divides its local dataset into support set and query set (half of each in our experiments). The support sets are utilized for learning how to solve the task, by performing R rounds of FL. The query sets are utilized for evaluating the performance on this task and performing the meta-update process. Class prototype is the mean of the model output for the data samples in a specific class. We will make these definitions clearer in the revised manuscript.
>
> 4. **Organization of Section 2.2.1:** To be precise, Algorithm 1 can be divided into two big steps: first is the “R rounds of local updates and aggregations” step (Section 2.2.1) and the second is the “one-round local meta-update and aggregation” step (Section 2.2.2). As the reviewer suggests, we will refer to the relevant lines in Algorithm 1 in each bolded section.
>
> 5. **Assumptions in Section 3**: In order to make the challenging analysis tractable, these assumptions are commonly adopted in FL literature [Li20], [Fallah20], [Lin20]. It is true that adversaries can make the variance larger in Assumptions 2 and 3. In such cases, although the analysis remains the same, we will observe a larger error term in equation (5) due to the increased bounds $V_d$ and $V_p$. To handle this issue, various adversary-mitigating schemes can be directly applied to our algorithm.
>
> [Li20] X. Li et al.., “On the convergence of FedAvg on non-IID data,” ICLR 2020.
>
> [Fallah20] A. Fallah et al., “Personalized federated learning with theoretical guarantees: A model-agnostic meta-learning approach,” NeurIPS 2020.
>
> [Lin20] S. Lin et al., “A collaborative learning framework via federated meta-learning,” arXiv preprint arXiv:2001.03229, 2020.
>
> 6. **Clarifying why the result in Fig. 3(c) is more desirable than Fig. 3(b):** In Fig. 3, diamonds represent data at a given client while circles indicate data from all other client. In both figures, the two classes at the given client (green and red diamonds) are well-separated. However, in Fig. 3(b), these two classes are close to magenta and blue circles (different classes at other clients). In comparison, in Fig.3(c), the green diamonds are close the green circles and the red diamonds are close to the red circles. What happened is that by considering the global prototypes (reflecting classes of all participants), the classes are trained to be far away from all other classes, which leads to a stronger classification.
>
> 7. **Typos:** We corrected all the typos accordingly.
>
> We again appreciate the reviewer for the time and efforts. Please do let us know if you have any further comments/questions.
>
> Best, Authors

---

> ### Author Response · Authors · 2021-08-29
> **Thanks for the update**
>
> We really appreciate your score update. We will definitely make the "downstream" target clarifications in the abstract as well as throughout the paper. Thanks again for the helpful suggestion.
>
> Best, Authors

---

### Official Review · Reviewer_1tsf · 2021-07-22

**Rating:** 7
**Confidence:** 3

**Summary:**

The paper introduces a problem that is based on both meta-learning and federated ideas.  They call this few-round learning, where the goal is to prepare an initial model that can quickly adapt to any group of clients within only a few rounds of FL. The problem is in my opinion significant and has a lot of practical application. The introduce a meta-learning algorithm using few rounds of FL followed by inference, to be performed by a group of clients on a possibly unseen task. The also show experiments that the scheme outperforms  existing  pre-training  approaches including fine-tuning via FedAvg and personalized FL in both IID and non-IID scenarios. I think the paper has core value for the FL field.

**Limitations And Societal Impact:**

I am not quite sure why there is a section comparing the results with the personalized FL while the goal here is obviously (and as stated in the paper) different. It would have been better if other FL methods apart from FedAVG were considered for the comparison instead of the focus on comparison with personalized FL. Personalized FL is in my opinion the opposite problem to the problem at hand so not sure of the value of the comparison although we see improvement on the personalization part.

A few papers worth adding in references and comparisons:

https://proceedings.neurips.cc//paper_files/paper/2020/hash/fb2697869f56484404c8ceee2985b01d-Abstract.html
https://openreview.net/forum?id=BkluqlSFDS



**Main Review:**

The paper reads well and contributes a new solution to a unique and interesting problem. The authors have explained the problem and the importance of it in the field if federated learning very well. The main contribution in my opinion is  the fact that they build a global model that can be re-used for several tasks with few learning episodes at the local clients. In their words, "The global prototypes serve as prior knowledge, a form of regularization, and prevent local models from overfitting to the local data. Moreover, the global prototypes (reflecting all classes across clients) can assist the local model to learn a more general embedding space". So basically we have a system that learns the general features of the data at the global layer and relies on local models for specific features in the local data. More over these local models can benefit from the meta-models to have better generalization behavior. I think authors can do a bit of a better prior work summarization on the FL field and introduce comparisons with other FL techniques that outperform FedAvg.




**Time Spent Reviewing:**

8

---

> ### Author Response · Authors · 2021-08-10
> **Response to Reviewer 1tsf**
>
> We thank the reviewer for their efforts as well as their valuable comments. Below, we address the comments that the reviewer raised.
>
> 1. **Comparison with personalized FL:** We agree with the reviewer’s point in that the direct comparison between our method and personalized FL does not serve the purpose well. We were just curious about how the “forced” global model constructed from personalized FL (with a strong meta-leaning flavor) would fair. We will move the comparison table (and discussion) between our scheme and personalized FL from the main manuscript to Supplementary Material.
>
>
> 2. **Comparison with other FL techniques that outperform FedAvg:** About using a better FL technique, as can be seen in Lines 13 and 21 of Algorithm 1, our algorithm adopts FedAvg when aggregating the models. Hence, for a fair comparison, we utilized FedAvg for the model aggregation process of all other baselines. We note that other aggregation methods that performs better than FedAvg (such as FedMA that the reviewer suggested) is applicable not only in other baselines but also in our scheme. Adopting other aggregation methods that outperforms FedAvg can further improve the performance of our method and other baselines. We will try to make this point clearer in the revised manuscript; thanks for the comment.
>
> Again, we appreciate the reviewer for the time and efforts. Please let us know if you have any other comments/questions.
>
> Best, Authors

---

### Official Review · Reviewer_NDpP · 2021-08-03

**Rating:** 6
**Confidence:** 4

**Summary:**

This paper studies a new problem setting called few-round learning in federated learning, and proposes a meta-learning based method to solve the problem. Experiments on real datasets are used to verify the effectiveness of the proposed method.

**Main Review:**

This paper studies a new problem setting called few-round learning in federated learning, and proposes a meta-learning based method to solve the problem. Experiments on real datasets are used to verify the effectiveness of the proposed method.

The few-round learning problem seems to be interesting and useful in federated learning, because it can reduce the cost in federated learning. The proposed meta-learning based method seems to be reasonable. Furthermore, the proposed meta-learning itself can be completed via federated learning. Experiments on real datasets show that the proposed method can outperform baselines for arbitrary groups of clients.

The experiments can be improved. The datasets used for evaluation are those for meta-learning evaluation which are relatively small, but this paper is for federated learning rather than meta-learning. Hence, adopting datasets suitable for federated learning can improve the convincingness of the proposed method. Furthermore, the paper does not introduce the hardware and platform of the experiments. Experiments on a platform of real applications (or similar settings) can also make the results more convincing.


**Time Spent Reviewing:**

6

---

> ### Author Response · Authors · 2021-08-09
> **Response to Reviewer NDpP**
>
> We appreciate the reviewer's comments on our paper. Our answers to the reviewer's comments are given as below:
>
> 1. **Comment on the datasets we used:** We appreciate the comment. To validate our approach further based on the reviewer’s comment, we performed additional experiments with a larger dataset, TinyImageNet. It contains 200 classes and 120000 image samples in total. We use 180 classes as train classes (for the meta-training phase) and remaining 20 classes as test classes (for the deployment phase). The performance was measured in a non-IID setting under the same setup in Table 2 of the main manuscript. As a result, the proposed FRL with GPAL achieves 65.43 ± 0.39%. Without GPAL performance degrades somewhat to 64.45 ± 0.41%. For fine-tuned FedAvg, accuracy sharply drops to 34.99 ± 0.38%. When there is no fine-tuning, FedAvg yields 28.54 ± 0.24%. As for personalized FL baselines, the distance-based classifier shows 57.66 ± 0.43% while the linear classifier shows 55.89 ± 0.42 %. After all, the experimental results in TinyImageNet show a consistent trend to the main paper, validating the superiority of the proposed few-round learning method. We also note that we already considered **FEMNIST**, one of the common benchmark datasets for federated learning, based on which our approach outperforms various baselines. Finally, we note that miniImageNet and CIFAR100, other datasets we utilized in this paper, have more classes than MNIST and CIFAR10, the datasets commonly adopted in federated learning.
>
> 2. **Hardware and platform:** All of our experiments is based on pytorch, and a single NVIDIA GeForce GTX 2080 Ti GPU is used. The code can be found in the Supplementary Material. Based on the reviewer’s comment, we will clarify these details in the revision.
>
> Thanks again for your time and efforts in reviewing our paper. Please let us know if there are any additional questions that we should address.
>
> Best, Authors

---

> > ### Comment · Reviewer_NDpP · 2021-08-28
> > **Thanks for the response**
> >
> > 1. About the datasets: Thanks a lot for providing more details about the datasets.
> > 2. It seems that the experiments are performed in a simulated way with only one machine (GPU), which is not a real federated artitecture.

---

> > > ### Author Response · Authors · 2021-08-28
> > > **Thanks for the reply**
> > >
> > > Thanks for the reply. As can be seen in our submitted code, we didn't utilize a real platform with distributed hardware, but implemented our federated learning setup using PyTorch and obtained results via simulations.
> > >
> > > Thank you,
> > >
> > > Authors

---

> > > ### Author Response · Authors · 2021-08-30
> > > **Additional comments to Reviewer NDpP**
> > >
> > > Please allow us to add a clarification regarding your comment on real federated architecture. When **comparing the training times** of different schemes, we agree that simulation with multiple machines (GPUs) could be more convincing, but we stress that we are **comparing classification accuracies after a fixed number of FL rounds**, in which case a single GPU versus multiple GPUs would yield exactly the same results. The best known works of the recent past, like FedAvg and FedProx as well as other frequently cited papers in FL (as listed below), also have resorted to the same method as ours. Once again, we appreciate your feedback.
> > >
> > > [FedAvg] H. B. McMahan et al., "Communication-Efficient Learning of Deep Networks from Decentralized Data," AISTATS 2017.
> > >
> > > [FedProx] T. Li et al., "Federated Optimization in Heterogeneous Networks," MLSys 2020.
> > >
> > > T. Li et al., "Fair Resource Allocation in Federated Learning," ICLR 2019.
> > >
> > > V. Smith et al., "Federated Multi-Task Learning," NeurIPS 2017.

---

### Decision · Program_Chairs · 2021-09-27

**Decision:**

Accept (Poster)

**Comment:**

This paper proposes a meta-learning approach for federated learning which allows arbitrary groups of clients can obtain a good global model in a few communication steps.

The reviews were quite positive and the author response (which contained additional experimental results) helped to further consolidate the scores. Overall, the problem of few-round FL and the proposed approach were deemed to be nicely motivated, interesting and useful in practice.

Therefore, the paper is accepted.